



# Impact of deoxygenation and hydrological changes on the Black Sea nitrogen cycle during the Last Deglaciation and Holocene

Anna Cutmore[1*]., Nicole Bale[1]., Rick Hennekam[2]., Bingjie Yang[1]., Darci Rush[1]., Gert-Jan Reichart[2,3]., Ellen C. Hopmans[2]., Stefan Schouten[1,3]

[1]Department of Marine Microbiology & Biogeochemistry, NIOZ Royal Netherlands Institute for Sea Research, 1790 AB Den Burg, Netherlands

[2]Department of Ocean Systems, NIOZ Royal Netherlands Institute for Sea Research, 1790 AB Den Burg, Netherlands

[3]Department of Earth Sciences, Universiteit Utrecht, Princetonlaan 8a, 3584 CB Utrecht, Netherlands

*Corresponding Author: anna.cutmore@nioz.nl

**Abstract**

The marine nitrogen (N) cycle profoundly impacts global ocean productivity. Amid rising deoxygenation in marine environments due to anthropogenic pressures, understanding the impact of this on the marine N-cycle is vital. The Black Sea's evolution from an oxygenated lacustrine basin to an anoxic marine environment over the last deglaciation and Holocene offers insight into these dynamics. Here, we generated records of the organic biomarkers heterocyte glycolipids, crenarchaeol, and bacteriohopanetetrol, associated with various water-column microbial N-cycle processes, which indicate a profound change in Black Sea N-cycle dynamics at ~7.2 ka when waters became severely deoxygenated. This transition substantially reduced Thaumarchaeota-driven nitrification and enhanced loss of bioavailable nitrogen through anammox. In contrast, other climatic changes over the last deglaciation and Holocene, such as freshwater input, water-level variations and temperature changes, did not impact these processes. Cyanobacterial nitrogen fixation in surface waters proved more responsive to changes in salinity and associated water column stratification. Our results indicate that future deoxygenation in marine environments may enhance bioavailable nitrogen loss by anammox and reduce nitrification by Thaumarchaeota, while enhanced stratification may increase cyanobacterial nitrogen fixation in the surface waters.

## 1. Introduction

The marine nitrogen (N) cycle is a significant control of biological productivity in our global oceans. It is directly connected to the fixation of atmospheric carbon dioxide and carbon export from the ocean's surface, influencing atmospheric $CO_2$ levels over geological time scales (Falkowski et al., 1998). As the marine N-cycle is strongly regulated by biology, the (de)oxygenation of the ocean determines the microorganisms involved in these biogeochemical cycles and the aerobic/anaerobic pathways that occur. Under anoxic conditions, loss of bioavailable nitrogen is substantial, attributed to anaerobic ammonium oxidation (anammox) and denitrification (Kuypers et al., 2003; Dalsgaard et al., 2012). With deoxygenation in marine environments increasing due to anthropogenic climate and environmental changes (i.e., Keeling et al., 2010; Bopp et al., 2013), and research linking deoxygenation to changes in the marine N-cycle (Kalvelage et al., 2013; Naafs et al., 2019), it is important



to enhance our understanding of how the marine N-cycle may respond to future deoxygenation and what the
associated feedbacks on carbon fixation might be.

Marine basins that have experienced changes in oxygenation in the past can provide perspective on the current
deoxygenation of modern global oceans and the associated feedbacks in the marine N-cycle, in particular on
timescales beyond the observational record. Today, the Black Sea is the world's largest permanently stratified
anoxic basin with limited connection to the global ocean through the Bosporus Strait and its redox gradient is a
hotspot of diverse microbial populations and metabolisms (Kusch et al., 2022). However, over the last
deglaciation and Holocene (approximately the last 20 ka), the Black Sea experienced large hydrological changes.
The basin was an oxygenated fresh-water lacustrine environment during the Last Glacial Maximum (LGM)
(Schrader, 1979) and experienced many environmental changes during the subsequent deglaciation, including
temperature changes (Bahr et al., 2005; 2008; Ion et al., 2022), water-level variations (Ivanova et al., 2007;
Nicholas et al., 2011; Piper & Calvert, 2011), and changes in freshwater input into the basin, both through
melting of Eurasian icesheets and alpine glaciers after the LGM and changes in regional precipitation (Bahr et
al., 2005; 2006; 2008; Badertscher et al., 2011; Shumilovskikh et al., 2012). It became reconnected to the global
ocean at ~9.6 ka when post-glacial sea-level rise caused an initial marine inflow (IMI) over the Bosporus sill (Aksu
et al., 2002; Major et al., 2006; Bahr et al., 2008; Ankindinova et al., 2019), leading to enhanced salinity of the
upper part of the water column (Marret et al., 2009; Verleye et al., 2009; Filipova-Marinova et al., 2013) and
euxinic deep waters developing in the basin after ~7.2 ka (Arthur & Dean, 1998; Eckert et al., 2013). Thus,
sedimentary records of the Black Sea may provide a unique perspective of the impact of deoxygenation, as well
as changing temperature and salinity, on the marine N-cycle.

Diagnostic lipid biomarkers of microbes preserved in the geological record can offer a unique insight into past
changes in the N-cycle (Rush & Sinninghe Damsté, 2017 and references cited therein; Elling et al., 2021; van
Kemenade et al., 2023). Nitrogen fixing heterocytous cyanobacteria play a crucial role in transforming nitrogen
gas ($N_2$) to bioavailable nitrogen ($NH_3$) and sustaining primary productivity in both marine and freshwater
environments (Villareal, 1992; Ploug et al., 2008). Identification of their diagnostic biomarkers, heterocyte
glycolipids (HGs), in the geological record enables exploration of past changes in nitrogen fixation by these
microbes (Bauersachs et al., 2009; 2010; Sollai et al., 2017; Bale et al., 2019; Elling et al., 2021). Nitrification, the
microbial two-step conversion of ammonia ($NH_3$) and/or ammonium ($NH_4^+$) to nitrate ($NO_3^-$), is a central part of
the marine N-cycle. Archaea of the phylum Thaumarchaeota (also known as Nitrososphaerota) are among the
most abundant and widespread marine prokaryotes (Karner et al., 2001; Francis et al., 2005), playing a crucial
role in nitrification in the Black Sea (Lam et al., 2007) by aerobically oxidizing ammonia to nitrite (Könneke et al.,
2005; Wuchter et al., 2006). As Thaumarchaeota are the exclusive producers of the membrane spanning lipid,
crenarchaeol (Sinninghe Damste et al., 2002), this biomarker can be used to identify Thaumarchaeota in the
geological record and explore the palaeo marine N-cycle. Another critical part of the N-cycle is the loss of
bioavailable nitrogen to $N_2$. Under anoxic conditions, bioavailable nitrogen ($NO_3^-$, $NO_3^-$, $NH_3$ and $NH_4^+$) can be
lost through two processes in subsurface waters: anammox (van de Graaf et al., 1997; Kuypers et al., 2003) and





denitrification (Kuenen and Robertson, 1988). It is possible to explore past changes in anammox activity in the
sedimentary record using the unique ladderane fatty acids (Sinninghe Damste et al., 2002) but these are
relatively poorly preserved in sediments (Jaeschke et al., 2007). Alternatively, the ratio of bacteriohopanetetrol
(BHT)-34S (which is ubiquitously synthesized by aerobic bacteria) and the later eluting stereoisomer BHT-x
(which is predominately synthesized by marine anammox bacteria, i.e., Ca. Scalindua spp.) (Rush et al., 2014;
Schwartz-Narbonne et al., 2020; van Kemenade et al., 2023) can be used to trace past anammox activity.
Denitrification is performed by a large range of organisms (Knowles, 1982), but at present, there are no
associated diagnostic lipid biomarkers (Rush et al., 2017).

In this study, we used lipid biomarkers of microbes involved in the N-cycle in combination with other
geochemical records from a sediment core located in the western Black Sea spanning the last deglaciation and
Holocene (~20 ka – present) to better constrain and assess the sensitivity of the marine N-cycle under changing
hydrological and oxygenation conditions and explore its potential links to broader global climate dynamics.

**2. Regional Setting**
The Black Sea is a large meromictic marginal basin connected to the Mediterranean Sea via the Turkish Straits
(the Bosporus, the Sea of Marmara, and the Dardanelles Strait) (Fig. 1). The Black Sea has a net outflow into the
Aegean Sea via the Turkish Straits, and is primarily supplied by three major rivers, the Danube, Dnieper, and
Don. With freshwater flowing out of the basin and dense, highly saline waters flowing in, the water column is
highly stratified with respect to salinity (density). An oxygenated colder surface layer (0 – 50 m) overlies warmer,
anoxic, sulfidic, hypersaline deep waters (100 – 2300 m), separated by a suboxic layer (50 – 100 m) (Murray et
al., 1989; 1995). The general circulation of Black Sea surface-waters is a basin-scale cyclonic boundary current
encompassing large eastern and western cyclonic gyres, with several smaller, anticyclonic coastal eddies (Fig. 1)
(Özsoy and Ünlüata, 1997).

**3. Methods**
During the cruise with the RV Pelagia in April 2017, piston core 64PE418 (235 cm length) was recovered from
1970 m below sea level (mbsl) depth in the Black Sea (42°56 N, 30°02 E) (Fig. 1). 44 sediment samples were taken
at 5 cm intervals along the depth of the core.

*3.1. Biomarker extraction and analysis*
Lipids were extracted from these samples using a modified Bligh and Dyer extraction method as described
previously (Bale et al., 2021). Using a mixture of methanol (MeOH), dichloromethane (DCM), and phosphate
buffer (2:1:0.8, v:v), the sediment was twice extracted ultrasonically (10 min). The combined supernatants were
phase-separated by adding DCM and phosphate buffer to create a solvent ratio of 1:1:0.9 (v:v). The organic
phase was collected, and the aqueous phase re-extracted three times using DCM. All extraction steps were then
repeated on the residue but with a mixture of MeOH, DCM and aqueous trichloroacetic acid solution (TCA) pH
3 (2:1:0.8, v:v). Finally, the organic extracts were combined and dried under a $N_2$ gas stream. A deuterated



betaine lipid {1,2-dipalmitoyl-sn-glycero-3-O-4'-[N,N,N-trimethyl(d9)]-homoserine; Avanti Lipids} internal
standard was added to each sample before filtering the extract through 0.45 µm cellulose syringe filters (4 mm
diameter; BGB, USA). Extraction blanks were performed alongside the sediment extractions, using the same
glassware, solvents and extraction methodology, but without sediment. Analysis of the extracts was performed
using the following UHPLC-HRMS reversed phase method. An Agilent 1290 Infinity I UHPLC was used, equipped
with thermostatted auto-injector and column oven, coupled to a Q Exactive Orbitrap MS with Ion Max source
with heated electrospray ionization (HESI) probe (Thermo Fisher Scientific, Waltham, MA). Separation was
achieved using an Acquity BEH C18 column (Waters, 2.1 × 150 mm, 1.7 µm) maintained at 30°C. The eluent
composition was (A) MeOH/H$_2$O/formic acid/14.8 M NH$_3$aq [85:15:0.12:0.04 (v:v)] and (B) IPA/MeOH/formic
acid/14.8 M NH$_3$aq [50:50:0.12:0.04 (v:v)]. The elution program was: 95% A (for 3 min) followed by a linear
gradient to 40% A (at 12 min) and then to 0% A (at 50 min), which was maintained until 80 min. The flow rate
was 0.2 mL min$^{-1}$. Positive ion HESI settings were: capillary temperature, 300°C; sheath gas (N$_2$) pressure, 40
arbitrary units (AU); auxiliary gas (N$_2$) pressure, 10 AU; spray voltage, 4.5 kV; probe heater temperature, 50°C;
S-lens 70 V. Lipids were analyzed with a mass range of $m/z$ 350–2000 (resolving power 70,000 ppm at $m/z$ 200),
followed by data-dependent tandem MS/MS (resolving power 17,500 ppm), in which the 10 most abundant
masses in the mass spectrum were fragmented successively. Optimal fragmentation was achieved with a
stepped normalized collision energy of 15, 22.5 and 30 (isolation width, 1.0 $m/z$) for IPL analysis (Bale et al.,
2021) and 22.5 and 40 (isolation width 1.0 $m/z$) for BHP analysis (Hopmans et al., 2021). The Q Exactive was
calibrated within a mass accuracy range of 1 ppm using the Thermo Scientific Pierce LTQ Velos ESI Positive Ion
Calibration Solution. During analysis, dynamic exclusion was used to temporarily exclude masses (for 6 s) to
allow selection of less abundant ions for MS/MS.

Biomarkers were identified based on their retention time, exact mass, and fragmentation spectra. Integrations
were performed on (summed) mass chromatograms of relevant molecular ions ([M+H]$^+$, [M+NH4]$^+$, and
[M+Na]$^+$) and in the case of crenarchaeol also the second isotope peak for each of the three adducts. Due to
coelution of BHT-34S, BHT-x isomer and an unknown nitrogen containing compound with the same mass,
identification and integration of BHT-34S and BHT-x was conducted using the $m/z$ 529.462 dehydrated insource
product ([M+H]$^+$-H$_2$O). Isoprenoidal glycerol dialkyl glycerol tetraether (isoGDGT) crenarchaeol, monohexose
crenarchaeol, and a crenarchaeol isomer were all integrated and combined as 'crenarchaeol'. The lipid
biomarker records are all presented as peak area per gram of total organic carbon (TOC).

***3.2. Total organic carbon and total nitrogen and $\delta^{15}N_{bulk}$ measurements***
Freeze-dried sediments were analysed for TOC, total nitrogen (TN) and bulk $\delta^{15}$N ($\delta^{15}N_{bulk}$) using a
ThermoScientific Flash EA Delta V Plus IRMS. Flow was 100 ml/min and the temperature for oxidation, reduction
and the oven were 900°C, 680°C, and 40°C, respectively. Nitrogen isotopic measurements were calibrated to
atmospheric air (AIR) and values are expressed in permil (‰) units. Inorganic carbon was removed from the
sediment prior to TOC analysis using HCl (2 mol), cleaned with bi-distilled water, then freeze-dried.



### 3.3. Age model

Accelerator Mass Spectrometry (AMS) $^{14}$C ages of bulk organic matter were measured from core 64PE418 (n =
7) to create a chronology on the 64PE418 depth scale. Samples were weighed and freeze-dried at NIOZ. The
AMS $^{14}$C measurements ($^{14}$C/ $^{12}$C) were determined using a Compact Carbon AMS System at the Poznań
Radiocarbon Laboratory, Poland. The sediment samples were pre-treated with 0.25M HCl (room temperature
overnight, then 80°C, 1+ hour), and rinsed with deionised water until pH = 7. Samples were then combusted in
closed (sealed under vacuum) quartz tubes, together with CuO and Ag wool (900°C, 10 hours). The $CO_2$ released
was then dried in a vacuum line and reduced with $H_2$ using 2 mg of iron (Fe) powder as a catalyst. The obtained
carbon and Fe mixture was then pressed into an aluminium holder (Czernik & Goslar, 2001). The measurement
was performed by comparing intensities of ionic beams of $^{14}$C, $^{13}$C and $^{12}$C measured for each sample and for
standard samples (with "Oxalic Acid II" used as modern standard; "coal" used as background standard of $^{14}$C-
free carbon). In each AMS run, 30-33 samples of unknown age were measured, alternated with measurements
of 3-4 samples of modern standard and 1-2 samples of background standard. The measured $^{14}$C/ $^{12}$C ratios are
corrected for isotopic fractionation and reported as conventional radiocarbon age according to Stuiver & Polach

167 (1977).


Seven bulk organic matter $^{14}$C dates were used in the production of the age-model for core 64PE418 (Table 1
and Fig. S3). Six of these were from this core, with an additional bulk organic carbon $^{14}$C date from the widely
acknowledged Unit I/II boundary of core KNR 134-08 BC17, which was used to further refine the age model for
the upper part of the core (Jones & Gagnon, 1994). Core KNR 134-08 BC17 was sourced from the same location
and water depth as 64PE418 and this boundary was identified in our core using the same significant colour and
elemental changes described in previous studies (Fig. S1 & S2) (i.e., Arthur & Dean, 1998; Bahr et al., 2005).
While seven $^{14}$C measurements were conducted on core 64PE418, one was excluded from the age model due to
an age reversal (142.5 cm), likely due to the presence of reworked material. Variable reservoir-ages were added
to our calibration (Table 1), using those calculated by Kwiecien et al., (2008) for intermediate water depths in
the Black Sea over the last deglaciation and Holocene. The $^{14}$C dates were calibrated using the Marine20
calibration curve (Heaton et al., 2020) for the upper three samples (24.5, 39, 76.5 cm) which reflect the period
after the infiltration of marine water; this is based on the colour and elemental changes in the core which
indicate that these samples fall within Units I and II (Arthur & Dean, 1998; Bahr et al., 2005). The lower four
samples (118.5, 158.5, 183.5 and 217.5 cm) were calibrated using the IntCal20 calibration curve (Reimer et al.,
2020), as they reflect the period prior to the marine infiltration when then Black Sea was a lacustrine
environment, as indicated by colour and elemental signatures in the core (Arthur & Dean, 1998; Bahr et al.,
2005). Using the R-code CLAM (Blaauw, 2010), the age–depth model was created based on the seven $^{14}$C dates.
Our age model shows that the 64PE418 biomarker records span the last 19.5 ka, with an average resolution of
~450 years. The following transitions are identified in our core by colour (Fig. S1) and elemental changes (Fig.
S2) and dated by our age model as follows: the onset of the IMI (138 cm) is at 9.6 ka ± 237 yrs, the boundary of
Unit II/III (96 cm) is dated at 7.2 ka ± 202 yrs, and the Unit I/II boundary (39 cm) is dated at 2.6 ka ± 402 yrs. The





dates of these boundaries align well with previously published calibrated ages for these transitions (i.e., Jones
& Gagnon, 1994; Ankindinova et al., 2019; Huang et al., 2021), as shown in Fig. S4.
**4. Results**
***4.1. TOC, TN and colour changes***
Sedimentary bulk TOC (%), bulk TN (%), and $\delta^{15}N_{bulk}$ (‰) range between 0.3 – 22.8% for TOC and 0.05 – 1.9% for
TN, and 5.2 – 0.0‰ for $\delta^{15}N_{bulk}$ (Fig. 2). There are significant colour changes in the core, as shown in Fig. S1 which
correspond to changes in TOC, TN and the elemental composition (Fig. S2). In the lower part of the core (19.5 –
9.6 ka), values are relatively low for TOC and TN, at ~0.84% and ~0.10%, respectively. At 9.6 ka, there is an
appreciable change in the elemental composition of the core, with increases in Ti/Ca, K and V and a decrease in
Mn/Al, which corresponds with a transition to darker sediments and an increase in TOC and TN to ~2.41% and
~0.26%, respectively. At 7.2 ka there is another major change in the colour and bulk elemental composition of
the core, with an increase in redox-sensitive elements U, V, and Mo and a decrease in Ti/Ca and K (Fig. S2), which
corresponds with darker sediments and increasing TOC values. TOC peaks between 6.6 – 4.6 ka (~21% for TOC
and ~1.7% for TN), declining towards the top of the core. $\delta^{15}N_{bulk}$ shows a general decline in values from the
upper to the lower part of the core. This decline is small between 19.5 – 7.7 ka (4.9 – 3.3‰), before a more
significant decrease to 1.2‰ at 6.6 ka (3.3 – 1.2‰). Values increase to 3.7‰ at 6.1 ka before declining to 0.0‰
at 3.9 ka, increasing slightly towards the top of the core to values of 1.3‰.
***4.2. Biomarkers***
We examined a number of lipid biomarkers related to the N-cycle in Black Sea core 64PE418 (Fig. S2). HGs were
identified in all samples (with the exception of 215 cm (16.4 ka)). These include HGs with a hexose ($C_6$) headgroup
i.e., hexose $C_{26}$ diol, hexose $C_{28}$ diol, hexose $C_{28}$ triol and hexose $C_{30}$ triol, which are specific to free-living
cyanobacteria, found in predominately freshwater and brackish environments (Bauersachs et al, 2009; Wörmer
et al., 2012). In addition, those with a pentose ($C_5$) headgroup i.e., pentose $C_{30}$ diol, pentose $C_{30}$ triol, pentose
$C_{32}$ triol were detected which are specific to cyanobacteria symbiotic with diatoms (diatom-diazotroph
associations, DDAs) (Schouten et al., 2013; Bale et al., 2015). Hexose HGs are present throughout the core,
increasing substantially in abundance between 9.6 – 6.6 ka, reaching maximum values at 9.6 ka. Pentose HGs
are detected from 4.3 ka onwards, increasing in abundance at the top of the record coinciding with low
abundance of hexose HGs. Crenarchaeol, a marker for Thaumarchaeota, was identified throughout our record,
showing high values in the early part of the record (~ 1.1E+14 peak area per g TOC) until 6.9 ka, abruptly shifting
to lower values ~ 3.9E+13 peak area per g TOC thereafter. The BHT-x ratio, a biomarker for anammox bacteria,
is low in the early part of our record (<0.3), due to low abundance of BHT-x. The BHT-x ratio increases after 6.9
ka to values around 0.3, due to higher abundance of BHT-x and lower abundance of BHT with a 34S
stereoconfiguration.





Finally, to reconstruct levels of oxygen in the subsurface waters of the Black Sea, isorenieratene was identified
(as described in Bale et al., 2021). Isorenieratene is a marker of the brown-coloured strains of the photosynthetic
green sulfur bacteria, Chlorobiaceae, which are anoxygenic photoautotrophs that require light and hydrogen
sulphide ($H_2S$); their presence indicates photic zone euxinia, whereby anoxic, sulfidic waters reached the photic
zone (Sinninghe Damste et al., 1993; Koopmans et al., 1996). Isorenieratene was identified in many of our
samples after 9.5 ka, peaking between 5.6 – 4.3 ka (reaching 3.39E+12 per g TOC at 5.6 ka), but was not detected
between 3.9 – 2.7 ka.

**5. Discussion**
Based on clear changes in TOC (Fig. 2), colour and elemental signatures (Fig. S1 & S2), we divided core 64PE418
into three widely acknowledged units, in line with previous studies (Jones & Gagnon, 1994; Arthur & Dean, 1998;
Bahr et al., 2005). Unit III spans ~20 – 7.2 ka, covering the period where the Black Sea was a lacustrine
environment, disconnected from the global ocean, and also the transition interval, where the basin moved
towards a marine environment after the IMI over the Bosporus sill at ~9.6 ka (Aksu et al., 2002; Major et al.,
2006; Bahr et al., 2008; Ankindinova et al., 2019). Unit II (~7.2 – 2.6 ka) and Unit I (~2.6 ka - present) span the
period where the Black Sea had become an anoxic brackish-to-marine environment.

**5.1. Oxic lacustrine phase (19.5 – 9.6 ka)**
Throughout the last deglaciation and early Holocene (19.5 – 9.6 ka), TOC and TN levels are low, likely due to
poor preservation of organic material, caused by the well-ventilated, oxygenated, freshwater environment that
existed in the basin at this time (Schrader, 1979). Isorenieratene is not detected during this period, while
elements that accumulate in sediment under anoxic conditions (i.e., Algeo and Li, 2020) also remained low (i.e.,
U, V, Mo; see Fig. S2), which all points to a well-oxygenated environment. Freshwater/brackish conditions
prevailed throughout this time, as shown by previous studies (Fig. S5; Filipova-Marinova et al., 2013; Ion et al.,
2022; Huang et al., 2022).Throughout this period, the abundance of Thaumarchaeota, indicated by crenarchaeol
abundance, and anammox, indicated by the BHT-x ratio, remained relatively steady. This stability is remarkable
since the region experienced significant climatic changes which led to large variations in the surface water
temperatures of the Black Sea, varying from ~10°C during the Bølling Allerød, ~7°C during the Younger Dryas
and ~14°C by the Early Holocene (Ménot & Bard, 2012), as well as changes in the input of freshwater into the
basin due regional precipitation variability and the melting of Eurasian icesheets and alpine glaciers (Bahr et al.,
2005; 2006; 2008; Badertscher et al., 2011; Shumilovskikh et al., 2012; Filipova-Marinova et al., 2013; Ion et al.,
2022). In contrast, changes in HG abundance and distribution suggest that surface-dwelling nitrogen-fixing
cyanobacteria were sensitive to hydrological changes in the Black Sea over this period (Fig. 3). The dominant HG
structure varies between hexose $C_{26}$ diol, hexose $C_{28}$ diol and hexose $C_{30}$ triol and after 11 ka, hexose $C_{28}$ triol
becomes present, which has been shown to be the major HG in members of the Rivulariaceae family (i.e.,
Calothix sp.) (Bauersachs et al., 2009). The warmer wetter conditions of the Early Holocene may have provided
a trigger for this change in HG abundance and composition. Indeed, an increase in the abundance of the
genus *Rivularia* was also noted in coastal regions of SW India during this period, coinciding with an increasingly



warm and wet climate (Limaye et al., 2017). Another cause for this shift may have been related to changes in
nutrient availability, with members of the Rivulariaceae family typically occurring in environments with highly
variable phosphorus availability (Whitton & Mateo, 2012).

**5.2. Transition phase (9.6 – 7.2 ka)**
In line with existing research (Arthur & Dean, 1998; Bahr et al., 2006; 2008), the IMI occurred at ~9.6 ka, leading
to a significant change in colour (Fig. S1) and elemental composition of the sedimentary record (Fig. S2), as well
as a substantial increase in abundance of HGs. This increase does not coincide with higher TOC content,
suggesting that enhanced preservation of HGs was not the cause. It is possible that these lipid biomarkers were
transported fluvially to this site from lakes within the catchment basin of the Black Sea due to the warm/wet
conditions at this time (Göktürk et al., 2011; Shumilovskikh et al., 2012; Filipova-Marinova et al., 2013). This,
however, appears unlikely as our site is located a substantial distance from the mouths of major rivers (>230
km), and the BIT index remains low during this period (~0.08; pers. comms. B.Yang), indicating only a minor
contribution of terrestrial organic matter at our site (Hopmans et al., 2004). Furthermore, as the proceeding
period (7 – 5.6 ka) was also warm and wet (Göktürk et al., 2011; Shumilovskikh et al., 2012; Filipova-Marinova
et al., 2013), we would expect the continuation of this peak if the HGs were being sourced from surrounding
lacustrine environments. Instead, these high values decline abruptly after 6.6 ka.

It is therefore likely that the peak abundance in nitrogen-fixing cyanobacteria is related to warmer Black Sea
surface temperatures during the early to mid-Holocene (Bahr et al., 2008) in combination with surface water
stratification (Bahr et al., 2006). This stratification may have been driven in part by enhanced freshwater influx
due to wetter conditions but may also have been triggered by the IMI through the Bosporus Strait at ~9.6 ka
(Major et al., 2006; Bahr et al., 2008; Ankindinova et al., 2019). This IMI likely led to the gradual salinisation of
the water column over this transition interval and intermittent build-up of anoxia in the water column. This, in
turn, led to periods of higher preservation of organic matter compared to the preceding period, as indicated by
the slight increase in TOC after 9.6 ka. The presence of isorenieratene after 9.4 ka indicates that anoxia reached
the photic zone at intermittent periods during this transition interval, thereby providing sufficient conditions for
the presence of the anoxygenic photoautotrophs, Chlorobiaceae. While the peak in nitrogen-fixing
cyanobacteria occurs ~2 ka before anoxia intermittently entered the photic zone, the initial influx of dense saline
water may have led to some reduction in vertical circulation, which reduced the amount of fixed nitrogen
upwelled to the upper water column, leading to the presence of nitrogen-fixing cyanobacteria at 9.6 ka. This
also coincides with a change in the distribution of HGs in our record between 9.7 – 6.9 ka where hexose $C_{28}$ diol
and hexose $C_{30}$ triol increase in abundance and hexose $C_{28}$ triol declines in relative abundance and is no longer
present after 9.1 ka, coinciding with the presence of isorenieratene. These changes may reflect a shift in species
composition, linked to the gradual salinisation and periodic anoxification of the water column after the IMI. The
IMI at ~9.6 ka appears, however, to have had little impact on the abundances of anammox and Thaumarchaeota.
This is possibly because basin-wide water column stratification and the permanent build-up of anoxia did not
occur until later in the record, meaning that neither process instantaneously reacted to the IMI at ~9.6 ka.



**5.3. Shift to anoxic brackish-to-marine mode of operation: a critical N-cycle threshold (~7.2 ka to present)**

After 7.2 ka there was a substantial increase in TOC and TN and an abrupt shift in parts of the subsurface N-cycle. The latter is shown by an increase in the BHT-x ratio, indicating an intensification of anammox, which is coeval with a decrease in crenarchaeol, indicating that there was a decline in Thaumarchaeota-driven nitrification. Studies have shown that by ~7.2 ka anoxia had built up in the water column, as indicated by changes in redox elements (Fig. S2 and Eckert et al., 2013; Wegwerth et al., 2018) and water column salinity had significantly increased (Fig. S5; Hiscott et al., 2007; Marret et al., 2009; Soulet et al., 2011; Filipova-Marinova et al., 2013), following the IMI from the Sea of Marmara at ~9.6 ka (Major et al., 2002; 2006; Bahr et al., 2005; 2008; Ankindinova et al., 2019). This is supported by the presence of isorenieratene in our record during this time, which indicates that anoxia penetrated the photic zone. This water column anoxia likely led to the enhanced preservation of TOC and TN and triggered a shift in the subsurface N-cycle, which crossed a threshold from an oxygenated lacustrine mode of operation to an anoxic brackish-to-marine mode of operation. The anoxic water column enabled anammox bacteria to expand their habitat from the anoxic sediments, where they likely were confined when the basin was an oxygenated freshwater environment, up into the suboxic/anoxic water column. This may therefore have commenced part of the modern-day N-cycle in the Black Sea where anammox activity occurs in the lower suboxic zone (~100 mbsl) where $O_2$ is (near) depleted and $H_2S$ is absent (Jensen et al., 2008), with anammox bacteria consuming ammonium diffusing from the deep sea and utilising the nitrite produced by both Thaumarchaeota and ammonia-oxidising bacteria (AOB) (Kuypers et al., 2003; Lam et al., 2007). Consequently, it may be that the abundance of anammox bacteria increased as a result of the coupling to nitrite production by other microbes in the suboxic zone, whilst benefitting from ammonium diffusing upwards from the deep sea. The increased anammox after 7.1 ka likely indicates that more bioavailable nitrogen was lost from the system after the switch to the anoxic brackish-to-marine mode of operation. At the same time, Thaumarchaeota abundance declined, which may be in part due to the build-up of anoxia in the water column which reduced the niche of these aerobic microbes and the nitrification performed by them. Once these processes crossed a threshold from an oxygenated lacustrine mode of operation to an anoxic brackish-to-marine mode of operation, they appear to have remained steady for the remainder of the Holocene despite changes in the salinity of the basin (van der Meer et al., 2008; Mertens et al., 2012; Coolen et al., 2013) and significant changes in regional temperature and precipitation (Göktürk et al., 2011; Shumilovskikh et al., 2012; Filipova-Marinova et al., 2013). This shows that deoxygenation was the main driver of the observed change in annamox as well as archaeal nitrification and that they were not affected by hydrological changes mainly occurring at the surface.

At 6.1 ka, the abundance of the HGs substantially declined, coinciding with an increase in $\delta^{15}N_{bulk}$, indicating a reduction in nitrogen fixation. As this decline in HG abundance and increase in $\delta^{15}N_{bulk}$ does not coincide with a reduction in TOC, it is unlikely that reduced preservation of HGs played a role here. As nitrogen-fixing cyanobacteria inhabit the upper surface layer, it is likely that this change is linked to the salinisation of the surface waters, with many studies demonstrating the disappearance of many freshwater mollusc, ostracod and





dinoflagellate cyst species at this time, which were replaced by an increased abundance of euryhaline
Mediterranean species (Hiscott et al., 2007; Marret et al., 2009; Filipova-Marinova et al., 2013; Ivanova et al.,
2015). At 6.1 ka, hexose $C_{26}$ diol and hexose $C_{28}$ diol are the only HGs present in the record, which may reflect
the dominance of genera in the Nostocaceae family (i.e., Anabaena sp., Aphanizomenon sp., Nodularia sp.,
Nostoc sp.), as these members demonstrate a dominance of the hexose $C_{26}$ diol and also contain varying
amounts of hexose $C_{28}$ diol (Gambacorta et al., 1999; Bauersachs et al., 2009). This distribution is similar to that
of the Baltic Sea after ~7.2 ka when a series of weak intrusions of saline water led to the basin becoming fully
brackish (Sollai et al., 2017). It is therefore possible that the peak in HGs in our Black Sea record between 9.6 –
6.9 ka represents a transition from the dominance of freshwater tolerant nitrogen-fixing cyanobacteria to more
brackish species, with brackish species dominating the surface-waters after 6.6 ka. After 6.1 ka, $\delta^{15}N_{bulk}$ gradually
decreases, indicating a rise in nitrogen fixation, as shown in previous studies (Blumenberg et al., 2009; Fulton et
al., 2012). It should be noted that a previous study has suggested, based on compound specific measurements
of pyropheophytin, that sedimentary $\delta^{15}N$ in the Black Sea is primarily derived from eukaryotic algae rather than
cyanobacteria (Fulton et al., 2012), meaning the use of $\delta15N_{bulk}$ as a nitrogen fixation signal must be used with
caution. HGs, however, are only derived from N-fixing cyanobacteria and are therefore an unambiguous
biomarker of nitrogen fixation. Interestingly, at 4.3 ka pentose HGs are detected, coinciding with lowest $\delta^{15}N_{bulk}$,
indicating the presence of marine nitrogen-fixing cyanobacteria found in symbiosis with marine diatoms. This
indicates that the surface water salinity had reached a threshold which enabled these marine microbes to
survive, with research indicating salinity reached ~17‰ during the deposition of Unit I (Ion et al., 2022) and
freshwater/brackish species had disappeared by this time (Fig. S5; Filipova-Marinova et al., 2013). Indeed,
reported increases in the number of euryhaline species at this time also points to the increasing salinity of the
surface waters (Marret et al., 2009; Bradley et al., 2012), which may be linked to warmer/drier conditions which
reduced freshwater influx and/or enhanced evaporation (Göktürk et al., 2011). Between 3.9 – 2.7 ka,
isorenieratene is not detected in the samples, reflecting the findings of previous studies (Sinninghe Damsté et
al., 1993). It has been suggested that this resulted from the erosion of the chemocline (Sinninghe Damsté et al.,
1993), while other research shows a short reoccurrence of freshwater/brackish species (Fig. S5; Filipova-
Marinova et al., 2013), which may indicate that enhanced freshwater input was responsible for lowering the
chemocline below the photic zone. The disappearance of hexose HGs after 0.6 ka indicates that surface water
salinities may more recently have become too high for the proliferation of brackish nitrogen-fixing
cyanobacteria.

## 6. Conclusions

This study shows a relatively stable subsurface N-cycle in the Black Sea over the last deglaciation and Holocene
with the exception of a critical threshold observed at 7.2 ka when the basin shifted from an oxygenated
lacustrine environment to an anoxic brackish-to-marine basin. At this time, the loss of bioavailable nitrogen
through anammox activity was enhanced and Thaumarchaeota-driven nitrification was reduced. Prior to, and
after this transition, the subsurface N-cycle was remarkably stable despite various climatic and hydrological
changes that impacted the basin during the deglaciation and Holocene periods. Both the amount of nitrogen





fixation by cyanobacteria and the composition of these microbes in the surface waters, however, appear to be
much more dynamic and sensitive to hydrological changes over this period, responding in particular to salinity
and temperature changes and stratification of the water column. Consequently, these records provide
important insight into how future deoxygenation and stratification in marine environments may affect the
microorganisms involved in the N-cycle. While deoxygenation in marine environments may lead to enhanced
loss of bioavailable nitrogen by anammox, and reduced nitrification by Thaumarchaeota, enhanced stratification
of the water column may lead to enhanced cyanobacterial nitrogen fixation in the surface waters. These changes
may have associated feedbacks on nutrient cycling and carbon fixation, with implications for the future global
carbon budget.

**Data Availability**
All data generated for this study are archived and publicly available via the Mendeley Data repository online at
https://10.17632/4c9fg7jf5d.1 (Cutmore et al., 2024).

**Acknowledgements**
We thank the Chief Scientist Prof. Laura Villanueva as well as the captain and crew of the *R/V* Pelagia for the
collection of core 64PE418. We would like to thank Jaap Sinninghe Damsté for useful discussions. For laboratory
support we thank Anchelique Mets, Denise Dorhout and Monique Verweij. Research cruise 64PE418 was funded
by the SIAM Gravitation Grant (024.002.002) from the Dutch Ministry of Education, Culture and Science (OCW).
This study was funded by the Netherlands Earth System Science Centre (024.002.001) from the Dutch Ministry
of Education, Culture and Science (OCW).

**Author Contributions**
Anna Cutmore: Conceptualization, Formal analysis, Investigation, Data Curation, Visualization, Writing - Original
Draft, Writing - Review & Editing; Nicole Bale: Conceptualization, Methodology, Investigation, Supervision,
Writing - Review & Editing; Rick Hennekam: Resources, Formal analysis, Investigation, Writing - Review & Editing;
Darci Rush: Formal analysis, Writing - Review & Editing; Bingjie Yang: Formal analysis, Investigation, Writing -
Review & Editing; Gert-Jan Reichart: Resources, Supervision, Writing - Review & Editing; Ellen C. Hopmans:
Supervision; Stefan Schouten: Conceptualization, Supervision, Funding acquisition, Writing - Review & Editing

**Competing interests:** The authors declare that they have no conflict of interest.

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



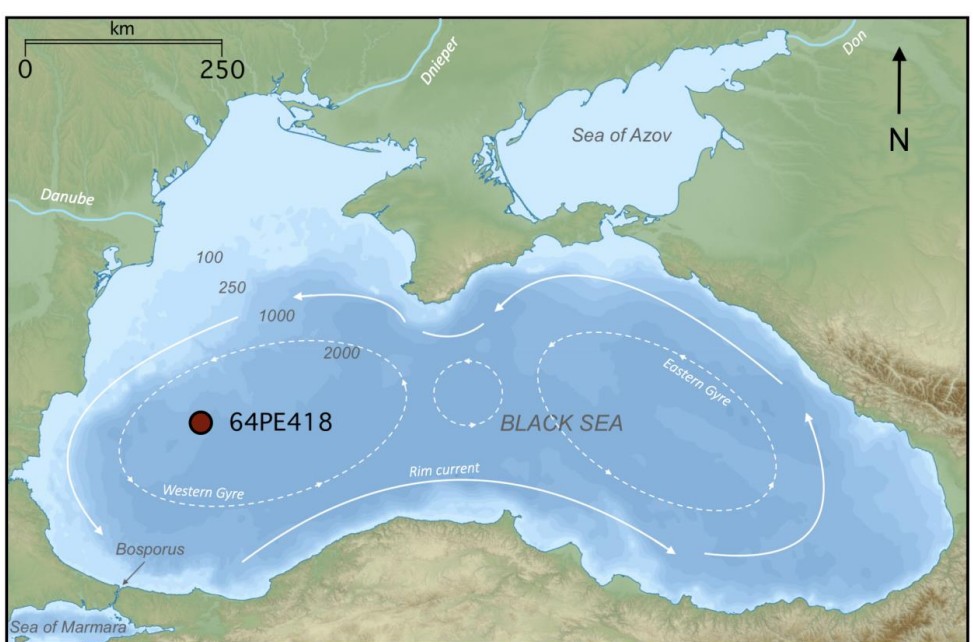

**Figure 1:** Map of the Black Sea basin, showing the major surface circulation and location of core 64PE418.
(Adapted from: Giorgi Balakhadze, English Wikipedia, 2016).



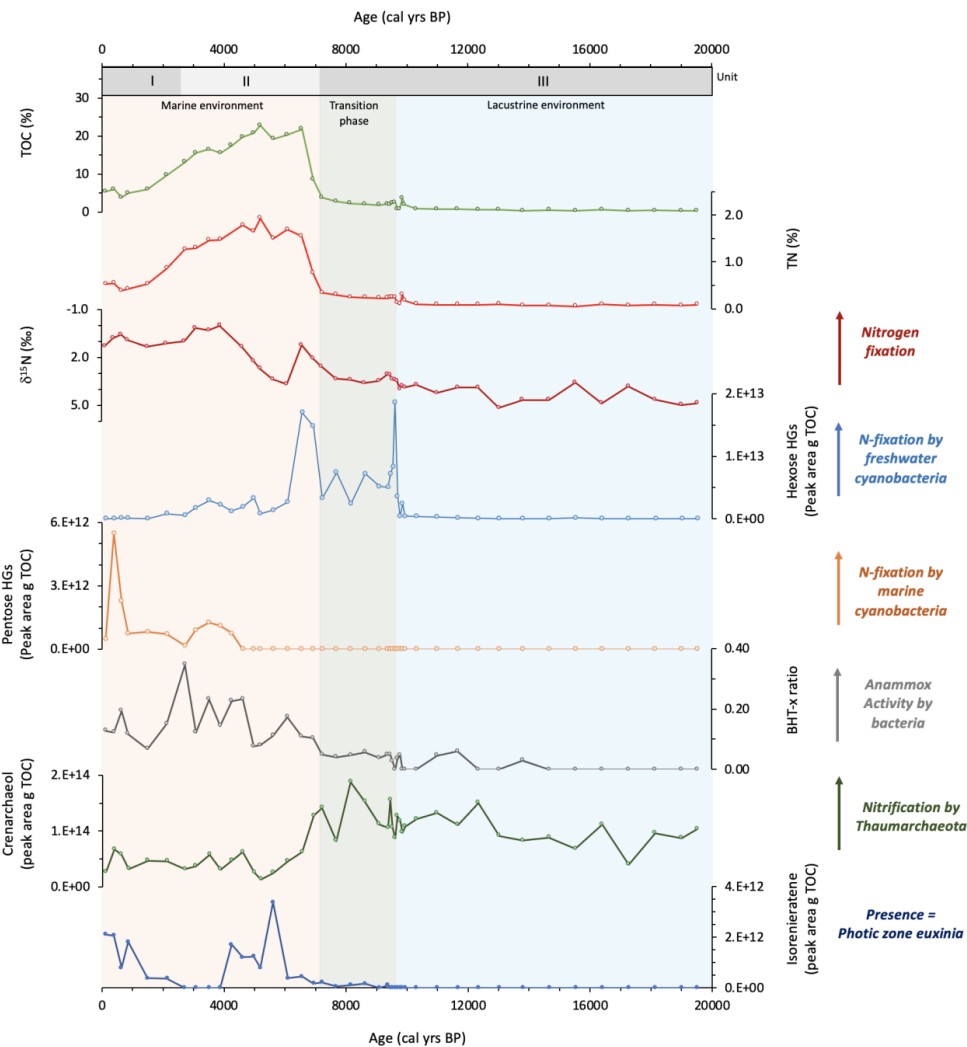

747

**Figure 2:** Geochemical records from Black Sea core 64PE418 of: a) TOC (%); b) TN (%); c) $\delta^{15}N_{bulk}$ (‰); d) hexose

749    HGs (peak area per g TOC); e) pentose HGs (peak area per g TOC); f) BHT-x ratio; g) crenarchaeol (peak area per

750    g TOC); h) isorenieratene (peak area per g TOC).



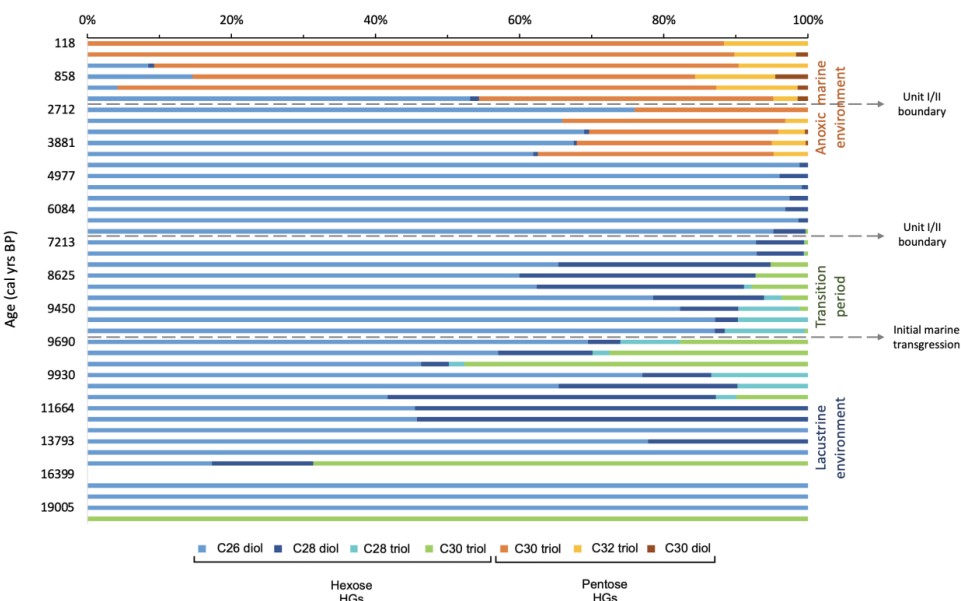

**Figure 3:** Changes over time in relative abundance of hexose and pentose HGs present in Black Sea core 64PE418

**Table 1:** Outline of the seven [14]C dates used in the production of the age-model for core 64PE418 and their calibrated ages. The [14]C and calibrated age of 142.5 cm is shown but was excluded from the age-depth model due to an age reversal.

| Core | Depth (cm) | Material | Radiocarbon age ([14]C yr BP) | ± 1σ | Calendar age (cal yr BP) | ± 2σ |
|---|---|---|---|---|---|---|
| 64PE418[a] | 24.5 | TOC | 2010 | 30 | 435[c,e] | 115 |
| KNR134-08-BC17[b] | 39.0 | TOC | 3640 | 70 | 2145[c,e] | 205 |
| 64PE418[a] | 76.5 | TOC | 5795 | 35 | 4870[c,e] | 170 |
| 64PE418[a] | 118.5 | TOC | 9110 | 50 | 9328[d,f] | 128 |
| *64PE418[a]* | *142.5* | *TOC* | *11650* | *60* | *12720[d,g]* | *50* |
| 64PE418[a] | 158.5 | TOC | 9670 | 50 | 9975[d,f] | 205 |
| 64PE418[a] | 183.5 | TOC | 12380 | 70 | 13358[d,g] | 123 |
| 64PE418[a] | 217.5 | TOC | 17420 | 110 | 19270[d,h] | 250 |

*a 14C dates from this study*
*b 14C dates from Jones & Gagnon, 1994*
*c Calibrated with the Marine20 curve (Heaton et al., 2020)*
*d Calibrated with the IntCal20 curve (Reimer et al., 2020)*
*e R-age of 600 years applied (Kwiecien et al., 2008)*
*f R-age of 800 years applied (Kwiecien et al., 2008)*
*g R-age of 900 years applied (Kwiecien et al., 2008)*
*h R-age of 1450 years applied (Kwiecien et al., 2008)*