# Peer review of "Impact of deoxygenation and hydrological changes on the Black Sea nitrogen cycle during"

_Climate of the Past, 2024_

## Author Response (AR1)

Reviewer 1,

Thank you for the detailed and helpful comments. Please find our point-by-point response to your comments below, outlining how we have addressed them in the manuscript.

1. The observed stability of the subsurface N-cycle is a key finding, but there is little discussion on what processes could have maintained this stability. The authors could consider elaborating on potential factors, such as community dynamics or nutrient availability, that might have contributed.

Thank you for this feedback. We have added a thorough discussion on the potential factors that maintained the stability of the subsurface N-cycle in our revised manuscript.

2. The use of hexose glycosides (HG) as indicators of cyanobacterial activity is somewhat speculative, especially given the shifts in salinity and environmental conditions throughout the study period. A more critical evaluation of this proxy's reliability would strengthen the interpretation. I suggest providing a more critical assessment of the assumption underlying biomarkers proxies, especially HGs.

Thank you for this suggestion. In the manuscript, we will provide a more critical evaluation of the use and assumptions of using HGs as an indicator of cyanobacterial activity.

3. The temporal resolution during critical transitions, particularly around 7.2 ka, may not be high enough to capture rapid or short-term variations in N-cycle dynamics. Acknowledging this limitation and its potential impact on the findings would be helpful

Thank you for this comment. We have acknowledged that there may be short-term changes in the N-cycle that are outside of the scope of this project but are of scientific interest for future work.

4. The reliance on modern analogues for interpreting past microbial and biogeochemical processes introduces uncertainties, especially given that past environmental conditions could differ significantly from present-day scenarios. This should be addressed more explicitly. For instance, the authors could expand the discussion a little bit to include a more nuanced view, highlighting the uncertainties of extrapolating modern findings to past conditions.

This is indeed a significant limitation of all proxies of environmental change, and we have made sure this important point is stated in the manuscript.

Thank you for the helpful detailed comments, we have addressed them all as follows:

Line 30: we have changed "our" to "the"

Line 35: we have added the complete anammox term to the abstract

Line 120-121: we have changed to "using heated..."

Line 216: we have clarified that these diatoms have a marine source

Figure comments:

We have moved the most mentioned S1 and S2 to the main text.

Reviewer 2,

Thank you for the comprehensive and helpful comments. Please find our point-by-point response to your comments below, outlining how we have addressed them in the manuscript.

**Major Comments**

1. **Implications for future ocean deoxygenation**: While the study provides valuable insights into the aquatic nitrogen cycle, it is highly unlikely that future ocean deoxygenation might lead to euxinic conditions similar to the Black Sea. There is no evidence of sulfidic waters in the modern global ocean, even in the most oxygen-depleted zones, such as the eastern tropical Pacific, which still contain significant nitrate concentrations. Furthermore, no model projections suggest that oxygen-deficient zones will evolve toward euxinia under global warming scenarios. The authors should conduct a thorough literature review and clarify this point to avoid potential confusion.

Thank you for this valuable comment. We have ensured that we are not suggesting that future global deoxygenation will always result in euxinia and have clarified that we are discussing N-cycle processes that are not specific to euxinia and that occur widely across various non-euxinic basins (i.e. nitrification by Thaumarchaeota, N-fixation by cyanobacteria and anammox by planctomycete bacteria), and are therefore not the result of euxinia, but influenced by changes in oxygenation of the Black Sea basin.

2. **Stratification-induced nitrogen fixation**: The authors conclude that stratification led to increased nitrogen fixation since 7.2 ka and suggest that future ocean stratification could similarly enhance nitrogen fixation. This extrapolation from the Black Sea to the global ocean is problematic. Stratification alone does not directly enhance nitrogen fixation. Rather, nitrogen fixation is more directly influenced by nitrogen limitation relative to phosphorus. In the modern Black Sea, intense nitrogen limitation, driven by anammox-induced nitrogen loss in the subsurface, is the key driver of surface nitrogen fixation. Enhanced stratification slows the upward supply of ammonium and promotes fixed nitrogen loss through anammox, indirectly increasing surface nitrogen limitation. A more detailed discussion connecting these paleo-data to modern Black Sea and global ocean observations is essential to contextualize these findings.

Thank you for this comment. We have expanded on this section to mention the importance of nitrogen limitation in promoting cyanobacterial N-fixation. We have also ensured that we are not extrapolating stratification and nitrogen fixation to the global ocean.

3. **Riverine nitrogen input**: Previous studies have suggested that riverine nitrogen input is a major source of fixed nitrogen in the modern Black Sea. The authors should discuss how riverine nitrogen input might influence their paleo-records and whether it impacts their interpretations of nitrogen cycle dynamics.

We have included references to riverine nitrogen input in the modern Black Sea and explained that it is unlikely to have influenced the Black Sea N-cycle over the Last Deglaciation and Holocene at our location to a large degree due to its remoteness from the coast.

4. **Comparison with existing d15N records**: The study presents a new $\delta^{15}N$ record, but comparisons with existing records (e.g., Fulton et al., 2012) are limited. A direct comparison of these $\delta^{15}N$ records, ideally plotted together, would highlight potential consistencies or discrepancies and strengthen the study. In addition, as the authors point out, bulk sedimentary $\delta^{15}N$ is well-known to reflect mixed signals; and its limitations should be discussed thoroughly.

Thank you for this suggestion. We have included a more thorough comparison of our d15N record to that of Fulton et al., 2012 and enhanced the discussion of the limitations of bulk d15N records.

5. **Diagenesis and preservation bias**: Bulk sedimentary $\delta^{15}N$ and biomarkers such as BHT-x and crenarchaeol are prone to diagenesis and preservation biases, which could complicate interpretations of microbial population dynamics. The authors should discuss how such biases may affect their results and the reliability of their conclusions.

We have addressed the possibility of diagenesis and preservation bias of the biomarkers in the Black Sea record, and think that it is unlikely to have played a major role in our records since large parts of the water column remained low in oxygenation and consequently organic carbon contents remained relatively high.

**Minor Comments**

Thank you for these suggestions, we have address them all as follows:

6. Line 23-24: We will specify the mechanisms by which salinity and stratification affect cyanobacterial nitrogen fixation to this section.

7. We agree that this is helpful information and have added modern water column chemical data (oxygen, nitrate, and ammonium profiles) in the Black Sea to the "Regional Setting" section.

8. Thank you for this suggestion, we have moved Table 1 to the supplementary material and the most referenced figures, S1 and S2, to the main manuscript.

9. Figure 1: We have added descriptions of depth contours

Dear Editor,

Thank you very much for this helpful feedback. We have added the suggested figures to the main document and addressed all the reviewer comments. Please find our point-by-point response to your comments, below.

A careful assessment of which figures remain key to the manuscript, rather than being as supplements, should be completed for the revised version. I agree with the requested revisions the reviewers request. I have some additional minor comments for consideration as you revise your manuscript:

1) you note in your age model (section 3.3) that there the sedimentation rate is not uniform through the sequence. Did you consider calculating biomarker fluxes to explore potential changes in export to the sediment? While this would indeed be useful, we do not have dry bulk density for the core, meaning it is not possible to calculate biomarker fluxes. Additionally, this age model does not allow the calculation of sediment flux on a high enough resolution.

2) also on the age model section: figure S3 shows that a date was discounted due to an age reversal. Can you explain why this date was discarded, beyond the reason given which was that the data point which follows is younger? If this discarded date was accepted you would have a relatively stable sedimentation rate from ~40-150 cm, then a shift in sediment rate below that i.e the depth at which the sedimentation rate changes would shift a bit earlier in time. I'm finding it difficult to see how these dates align with the stratigraphy, to see if the sedimentology changes align with two dates which could be problematic.

Thank you for this point, we have expanded on this in the manuscript. Due to the colour and elemental changes in the core, we have a good constraint on where key events occurred. If we included this date in the age model, the Unit III boundary would have been significantly older and out of line with the published dates of this boundary (around 9.6 ka) as shown in Fig. S4, therefore this date was excluded, and the date at 172.5 cm was included, as this produced an age model where the key events were in line with previously research.

3) age model comparison to other studies: Figure S4 compares this new work with previous studies. I agree that for Unit I/II there is good correspondence (lines 189-190) but there seem to be quite large differences for the other unit boundaries. Is this because they were calibrated with different reservoir ages, calibration programs, or something else?

Thank you for this comment. Attempts to decipher the chronology of Quaternary Black Sea deposits are hindered by a number of issues. Firstly, there is a lack of material adequate for dating, therefore these studies use different materials such as bulk TOC and bivalve shells. Secondly, there is significant inconsistency in the application of reservoir ages– some apply modern reservoir ages to the entire record, while others apply reservoir ages according to Soulet et al., 2011, Yanchilina et al. 2017 or Kwecien et al., 2008. Those of Kwecien et al., 2008 are most appropriate for our site as this paper published reservoir ages for intermediate water depths. We have added to the manuscript justification of our choice of reservoir correction and explained the complexity of dating Black Sea cores.

4) line 252-253 discusses abundances being relatively stable: this relates to my comment about calculating fluxes

noted above. Is the same pattern seen if fluxes are calculated, or is the description here referring to ratios or other independent measures which would not be affected by sedimentation rate?

The BHT-x record is a ratio and therefore is not affected by sedimentation rate. Fluxes aren't able to be calculated for the crenarchaeol record, but despite some slight changes in sedimentation rates over the oxic lacustrine period, crenarchaeol abundances do not appear to be significantly affected.

5) line 272-273: is the substantial increase in abundance of HGs due to the kink in the age-depth model? You note that TOC content doesn't change, which suggests that this might not be an issue, but it might be worth a double check.

While the sedimentation rate at this time is lower, there is not a coinciding rapid increase in TOC, which suggests that the HG abundance increase is not a result of this lower sedimentation rate. Additionally, this significant increase is not reflected in other proxies as would be expected if sedimentation was the main driver of this HG increase at this time.

Please let us know if you have any further questions or concerns.

Best wishes,

Anna Cutmore and the authors of manuscript CP-2024-59